# TRIM26 alleviates fatal immunopathology by regulating inflammatory neutrophil infiltration during *Candida* infection

Guimin Zhao[1,2], Yanqi Li[1,2], Tian Chen[1,3], Feng Liu[1,2], Yi Zheng[1,2], Bingyu Liu[1,2], Wei Zhao[1,3], Xiaopeng Qi[4], Wanwei Sun[1,2]*, Chengjiang Gao [1,2]*

**1** Key Laboratory of Infection and Immunity of Shandong Province & Key Laboratory for Experimental Teratology of Ministry of Education, Shandong University, Jinan, Shandong, P.R. China, **2** Department of Immunology, School of Basic Medical Sciences, Shandong University, Jinan, Shandong, P.R. China, **3** Department of Pathogenic Biology, School of Basic Medical Sciences, Shandong University, Jinan, Shandong, P. R. China, **4** Advanced Medical Research Institute, Shandong University, Jinan, Shandong, P. R. China

* sunwanwei2021@sdu.edu.cn (WS); cgao@sdu.edu.cn (CG)

**Data Availability Statement:** All relevant data are within the manuscript and its Supplemental Information files.

## Abstract

Fungal infections have emerged as a major concern among immunocompromised patients, causing approximately 2 million deaths each year worldwide. However, the regulatory mechanisms underlying antifungal immunity remain elusive and require further investigation. The E3 ligase Trim26 belongs to the tripartite motif (Trim) protein family, which is involved in various biological processes, including cell proliferation, antiviral innate immunity, and inflammatory responses. Herein, we report that Trim26 exerts protective antifungal immune functions after fungal infection. *Trim26*-deficient mice are more susceptible to fungemia than their wild-type counterparts. Mechanistically, Trim26 restricts inflammatory neutrophils infiltration and limits proinflammatory cytokine production, which can attenuate kidney fungal load and renal damage during *Candida* infection. Trim26-deficient neutrophils showed higher proinflammatory cytokine expression and impaired fungicidal activity. We further demonstrated that excessive neutrophils infiltration in the kidney was because of the increased production of chemokines CXCL1 and CXCL2, which are mainly synthesized in the macrophages or dendritic cells of *Trim26*-deficient mice after *Candida albicans* infections. Together, our study findings unraveled the vital role of Trim26 in regulating antifungal immunity through the regulation of inflammatory neutrophils infiltration and proinflammatory cytokine and chemokine expression during candidiasis.

## Author summary

Invasive fungal infections are one of the leading causes of death in immunocompromised patients. Concurrently, the emergence of drug-resistant pathogenic fungi poses a serious threat worldwide. Therefore, a better understanding of the mechanisms that protect the host from fungal infections is crucial. In the present study, we provide evidence that Trim26 plays a protective role in the antifungal immune response. Trim26 limits the

**Funding:** This work was supported by grants from the National Natural Science Foundation of China (82321002, 32230033, and 81930039 to CG.), grants from the National key research and development program (2021YFC2300603 to CG.), grants from Natural Science Foundation of Shandong Province (ZR2020MH165 to GZ.), and the grants from Cutting Edge Development Fund of Advanced Medical Research Institute awarded to YZ. The funders had no role in study design, data collection, and interpretation, or submitting the work for publication.

**Competing interests:** The authors have declared that no competing interests exist.

neutrophils infiltration into the kidneys, which is essential for alleviating kidney immuno-pathology following *Candida albicans* infection. Moreover, we found that excessive neutrophils infiltration into the kidney and aggravated renal immunopathology in Trim26-deficient mice were attributable to the elevated level of chemokine CXCL1 secreted by macrophages or dendritic cells after *C. albicans* infection.

## Introduction

*Candida albicans* is an important opportunistic human fungal pathogen that invades the mucosa and reach bloodstream, causing systemic fungal infections or invasive candidiasis. It is the third most common healthcare-associated infection with a high mortality rate (~40%) [1]. Most current antifungal drugs are ineffectual and produce adverse effects. Simultaneously, resistance to antifungal drug is rapidly increasing [2, 3]. Therefore, it is imperative to understand the immune response to *C. albicans* infection. A deeper understanding of host antifungal immunity is crucial for providing preventive guidance and developing novel treatment strategies to combat candidiasis [4].

During fungal infection, C-type lectin receptors (CLRs) play a critical role in the detection and recognition of fungal species such as *Candida* [4–6]. CLRs include dectin-1, dectin-2, dectin-3, Mincle, and mannose receptor C-type 1 (MRC1), which are mainly expressed in neutrophils, macrophages, and dendritic cells (DCs). Upon recognition of various components of the fungal cell wall, including β-glucan, α-mannan, hyphal mannose, and glycolipids, CLRs initiate antifungal immune responses [7–11]. In addition to the CLRs, toll-like receptor 2 (TLR2) is also involved in fungal recognition and activation of protective immunity [12]. Upon recognition of the respective ligands by various CLRs, spleen tyrosine kinase (SYK) is phosphorylated or activated. This is followed by the activation of CARD9–BCL10–MALT1 (CBM) complex-dependent nuclear factor-κB (NF-κB) and the mitogen-activated protein kinase (MAPK) signaling pathways, and subsequent release of proinflammatory cytokines and chemokines, including interleukin-6 (IL-6), IL-1β, IL-23A, and CXCL1. Th17 cell polarization and the subsequent release of IL-17A are then induced, which plays a vital role in neutrophil recruitment for fungal clearance [13–17]. However, the excessive production of proinflammatory cytokine, such as IL-6, IL-1β and TNF-α, may trigger an inflammatory storm and induce organ damage [18]. Following *C. albicans* hypha formation and invasion, neutrophils and mononuclear phagocytes are rapidly recruited to the infection site to mediate pathogen clearance via various antifungal responses [19]. The kidney is the main target organ involved in disseminated candidiasis. Nonetheless, the unrestrained recruitment of myeloid cells to the infected kidney and excessive production of inflammatory mediators accompanying this process often provoke an immune-driven pathology, leading to kidney damage, sepsis, and even death [20–24].

The E3 ligase Trim26 belongs to the tripartite motif (Trim) protein family and is involved in various biological processes, including cell proliferation, antiviral innate immunity, and the inflammatory response [25–29]. We previously demonstrated that Trim26 negatively regulates type I IFN signaling during antiviral immunity and positively regulates inflammatory immunity. This indicates that Trim26 plays a critical role in orchestrating specific immune responses based on the stimuli encountered [28, 29]. However, the involvement of Trim26 in the regulation of antifungal immune responses has not yet been established. In the present study, we found that Trim26 attenuates renal damage by restricting inflammatory neutrophil infiltration and limiting the production of proinflammatory cytokines in the kidney during *Candida* infection. We also provided evidence that Trim26-deficient neutrophils have a stronger

inflammatory phenotype and impaired fungicidal activity. Bone marrow-derived macrophages (BMDMs) and bone marrow-derived dendritic cells (BMDCs) that produce excessive chemokines, such as CXCL1, are responsible for the unrestrained recruitment of neutrophils. Taken together, our findings demonstrate a novel mechanism underlying Trim26 in protective antifungal immunity by controlling inflammatory neutrophil infiltration into the kidneys during fungemia.

## Results

### *Trim26^-/-^* mice were susceptible to systemic *C. albicans* infection

Previously, we found that Trim26 plays an essential role in the regulation of innate antiviral immune and inflammatory immune responses [28, 29]. In the present study, we investigated the role of Trim26 in the CLR signaling pathway and antifungal immunity. To test the potential role of Trim26 in antifungal immunity, we first challenged *Trim26^−/−^* mice and *Trim26^+/+^* littermates systemically with lethal doses of *C. albicans*. We found that *Trim26^−/−^* mice were more susceptible to *C. albicans*-induced fungal sepsis than wild-type (WT) control mice, which was manifested by a higher mortality rate, quicker weight loss, and higher fungal burden in the kidney, spleen, and liver of Trim26-deficient mice (Fig 1A–1C). Histopathologic analysis of kidneys also demonstrated that *Trim26^−/−^* mice exhibited increased renal inflammation and fungal burden (Fig 1D). Moreover, neutrophils infiltration into the kidney of the infected *Trim26^−/−^* mice was significantly higher than that of *Trim26^+/+^* mice (Fig 1D). To test whether Trim26 deficiency has any effect on immune cell development, we assessed the immune cell abundance in the spleen, bone marrow (BM), thymus, and lymph nodes in WT and *Trim26^−/−^* mice that were not infected with *C. albicans*. Trim26 knockout did not cause any abnormalities in the total cell number (S1A Fig) or the cell composition of immune cells in these organs. Specifically, the percentages of neutrophils (CD11b^+^-Ly6G^+^), monocytes (CD11b^+^-Ly6C^+^), macrophages (CD11b^+^-F4/80^+^), DCs (CD11c^+^-MHC II^+^), and NK cells (NK1.1^+^-CD3^−^) were similar between *Trim26^+/+^* and *Trim26^−/−^* mice (S1B and S1C Fig). The T-cells population in the thymus and lymph nodes was also unaffected by Trim26 deficiency (S1D and S1E Fig). These data suggested that Trim26 is dispensable for the development of innate and adaptive immune cells. Trim26 has been reported to be ubiquitously expressed in most of cell types [28]. To explore whether myeloid cells or tissue cells expressing Trim26 play a major role in antifungal immunity, BM-chimeric mice were generated by reconstituting lethally irradiated WT mice with syngeneic Trim26 KO BM or Trim26 KO mice with WT BM. WT mice reconstituted with Trim26 KO hematopoietic cells showed a phenotype similar to that of mice with total Trim26 deficiency in response to *C. albicans* infection (Fig 1E and 1F, *Trim26^−/−^→Trim26^+/+^* versus *Trim26^+/+^→Trim26^+/+^*). Furthermore, Trim26 KO mice reconstituted with Trim26 WT hematopoietic cells showed a phenotype similar to that of WT mice in response to *C. albicans* infection (Fig 1E and 1F, *Trim26^+/+^→Trim26^-/-^* versus *Trim26^-/-^→Trim26^-/-^*). These data indicated that hematopoietic cell-expressed Trim26 plays a major role in the antifungal immunity. Taken together, these results suggested that Trim26 positively regulates the antifungal immune responses *in vivo*.

### *Trim26^−/−^* mice showed stronger inflammatory damage following *C.albicans* infection

Although a host immune response is required to eliminate *C. albicans* infection, excessive inflammatory responses can lead to serious collateral tissue damage. Therefore, we determined the cytokine response in the kidneys of *Trim26^+/+^* and *Trim26^−/−^* mice following *C. albicans*

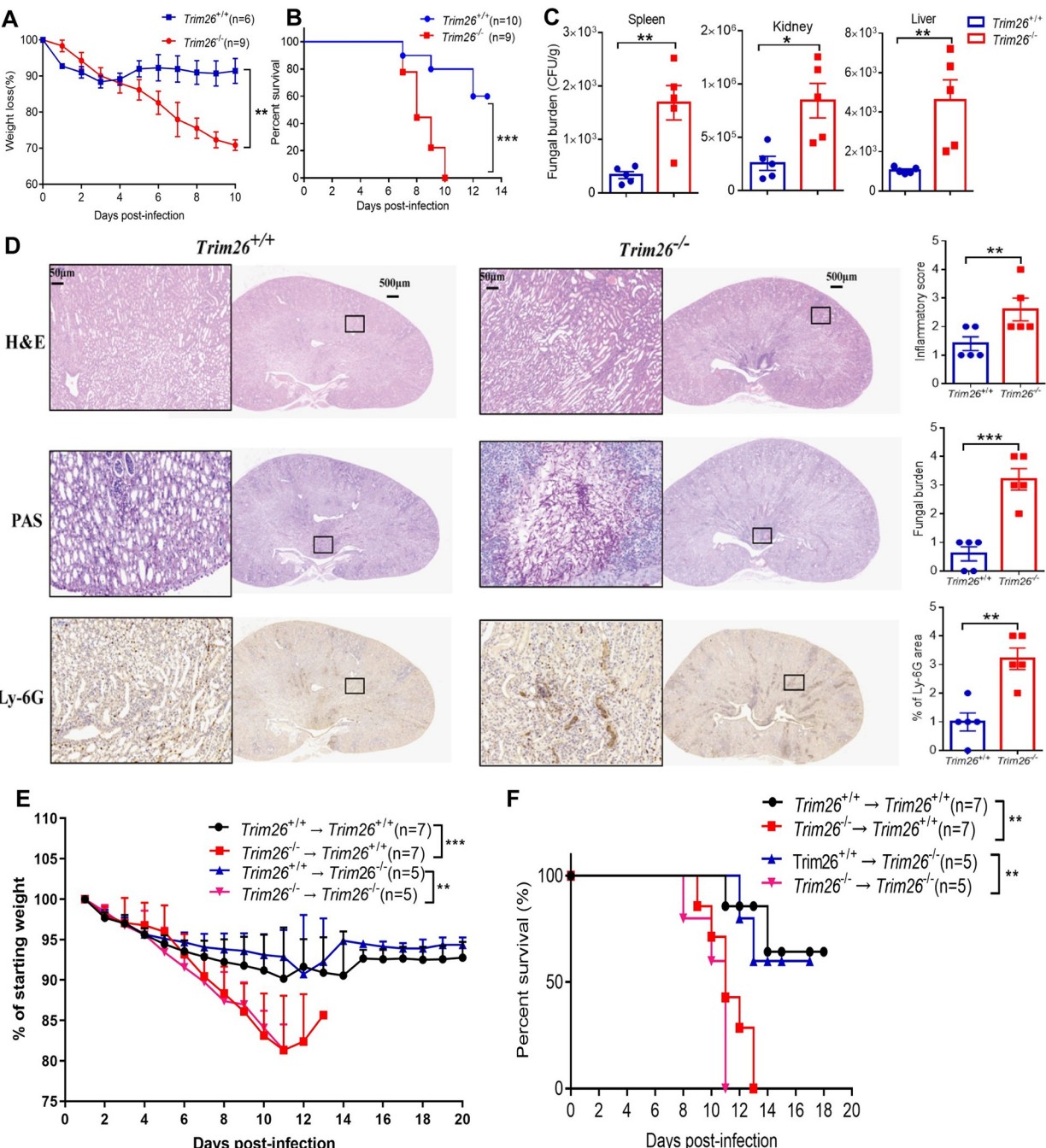

**Fig 1. Trim26 deficiency impaired anti-fungal immune responses *in vivo*.** (A-C) Wild-type control mice or Trim26-deficient mice were intravenous injected with live *C. albicans* ($2 \times 10^5$ cfu/100μL 1×PBS), and mice weight loss (A), survival curve (B) and the fungal burden in the spleen, kidney and liver (C) after 5 days post-infection were shown. (D) Kidney sections from *C. albicans*-infected *Trim26*$^{+/+}$ and *Trim26*$^{-/-}$ mice at day 5 were stained with hematoxylin and eosin (H&E), periodic-acid-Schiff (PAS) or Ly-6G (neutrophil marker). Insets show regions of renal inflammation, fungal growth and neutrophil infiltration, respectively, and are higher-magnification views of the boxed areas. Representative images of at least five replicates were shown. Scale bars, 500 μm (50 μm in insets). (E-F) BM-chimeric mice were established by reconstituting lethally irradiated WT mice with syngeneic Trim26-KO BM or Trim26-KO mice with WT BM as described in Materials and Methods. Mice were intravenously injected with live *C. albicans* ($2 \times 10^5$ cfu/100 μL 1×PBS), and the mice weight loss (E) and survival curve (F) were shown. *: $P < 0.05$; **: $P < 0.01$; ***: $P < 0.001$ based on two-tailed unpaired t-test (C-D), two-way ANOVA (A and E) and Log-rank (Mantel-Cox) test (B and F). Data are shown as mean ± SD. In A, B, one representative experiment of four independent experiments is shown. Each point represents a single mouse (C-D). In C, D, one representative experiment of three independent experiments is shown. In, E, F, one representative experiment of two independent experiments is shown.

infection. We found $Trim26^{-/-}$ mice had higher levels of proinflammatory cytokine expression than $Trim26^{+/+}$ mice, including IL-6, IL-1β, and TNF-α at day 5 post-infection (Fig 2A). Moreover, the expression of the anti-inflammatory factor IL-10 was significantly reduced in $Trim26^{-/-}$ mice compared with that in $Trim26^{+/+}$ mice (Fig 2A). To exclude the possibility that the change in cytokine expression was caused by differences in fungal burden, we analyzed cytokine expression levels when fungal abundance was similar. We first examined the fungal burden from day 1 to day 7 after *C. albicans* infection and found that the fourth day was the time point when the fungal burden was similar in both mouse types (S1F–S1H Fig). Consistent with the results shown in Fig 2A on day 5, the expression levels of IL-6, IL-1β, and TNF-α were higher in $Trim26^{-/-}$ mice at 4 days post-infection (Fig 2B). Furthermore, $Trim26^{-/-}$ kidneys showed significantly up-regulated expression of kidney injury molecule-1 (KIM-1) and neutrophil gelatinase-associated lipocalin (NGAL) (Fig 2C), markers of tubular epithelial damage used for the early diagnosis of acute kidney injury (AKI) [30–32]. We also examined the expression level of other inflammation mediators involved in acute kidney injury, including the major adhesion molecules ICAM-1 and P-selectin [33]. In agreement with the excessive inflammatory phenotype, the expression levels of these two adhesion molecules expression were significantly enhanced in $Trim26^{-/-}$ mice compared with their corresponding expression levels in $Trim26^{+/+}$ mice (Fig 2C).

Moreover, $Trim26^{-/-}$ kidneys showed severe swelling and pallor in gross pathology and exhibited a more pronounced weight gain after *C. albicans* infection than $Trim26^{+/+}$ kidneys (Fig 2D and 2E). To test whether the observed damage to renal architecture also impaired kidney function, we measured serum urea and creatinine levels 5 days after *C. albicans* infection. Consistent with the above results, the levels of serum urea and creatinine in $Trim26^{-/-}$ mice were much higher than those in $Trim26^{+/+}$ mice. These further indicated that $Trim26^{-/-}$ mice had more severe impairment of kidney function after *C. albicans* infection (Fig 2F). These results are consistent with previous findings that high levels of IL-6, IL-1β, and TNF-α produced in the kidney contribute to the immunopathogenesis of disease after systemic *C. albicans* infection [20, 30–31]. Taken together, our results suggested that uncontrolled fungal proliferation and excessive fatal proinflammatory cytokines production in $Trim26^{-/-}$ mice might cause severe renal inflammation, leading to fatal immunopathology.

## Trim26 deficiency expanded renal inflammatory neutrophil infiltration via increasing CXCL1 production during candidiasis

After access to the bloodstream, *C. albicans* can reach the glomeruli and penetrate the renal interstitium through blood vessels, then elicit an influx of inflammatory myeloid cells, which are capable of causing severe immunopathology and fatal kidney damage [20, 22]. To assess the effect of Trim26 on specific myeloid cell infiltration in the mouse *C. albicans* infection model, we harvested leukocytes from the kidneys and determined their immunophenotypes using FACS on days 1, 2, 3, 4, 5, and 7 post-infection. It was obvious that the neutrophil infiltration into the kidneys of $Trim26^{-/-}$ mice was significantly expanded, as shown by both neutrophils percentage and absolute number, compared with the corresponding parameters in $Trim26^{+/+}$ mice (Fig 3A–3C). These results were consistent with those of Ly-6G staining and mRNA expression in $Trim26^{-/-}$ mouse kidneys (Figs 3D and 1D). We also verified that the increasing of neutrophil infiltration was not attributable to neutrophil death (S2A Fig). However, no differences in macrophage accumulation were observed between $Trim26^{+/+}$ and $Trim26^{-/-}$ mouse kidneys after *C. albicans* infection (S2B and S2C Fig).

Myeloperoxidase (MPO) is a heme-containing peroxidase that is mainly expressed in neutrophils. After neutrophil activation, most of the MPO is sequestered into the phagosomes;

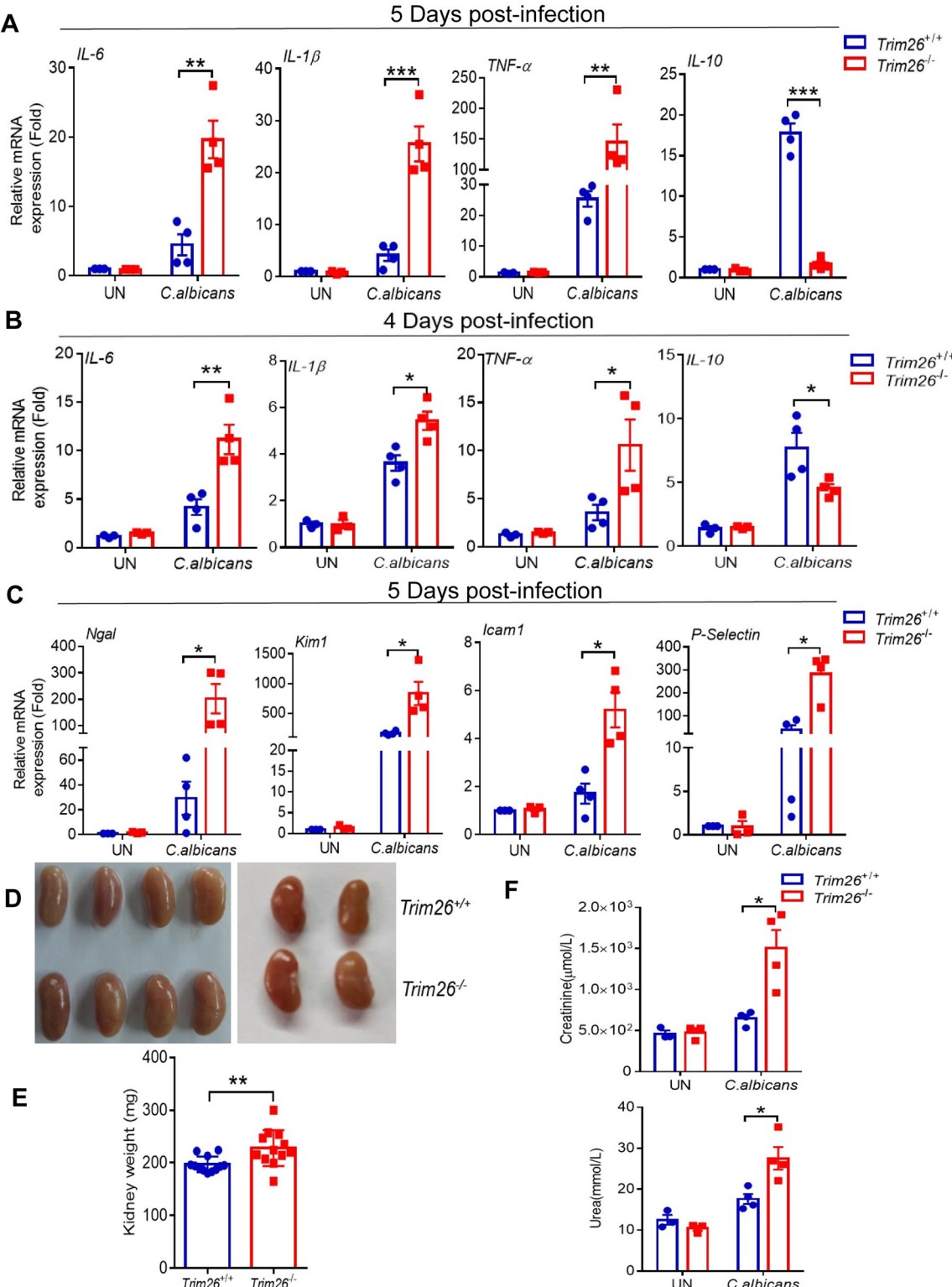

**Fig 2. *Trim26^-/-* mice showed stronger inflammatory damage following *C. albicans* infection.** WT mice or Trim26 KO mice were intravenously injected with live *C. albicans* (2×10⁵ cfu/100μL 1×PBS), and mice were euthanized at the indicated day after infection. Mice kidney was isolated and followed by real-time PCR analysis of indicated proinflammatory genes expression at day 5 after infection (A) and at day 4 after infection (B). Mice were euthanized at day 5 after infection. Kidney was isolated and followed by real-time PCR analysis of indicated acute kidney injury genes or adhesion molecules expression (C). (D and E) The weight of kidney after live *C.*

*albicans* infecting WT mice and Trim26 KO mice at five days. (F) Serum was isolated from the WT mice or Trim26 KO mice infected as in (A). Urea and creatinine concentration were measured by ELISA analysis. *: P<0.05; **: P<0.01; ***: P<0.001 based on two-tailed unpaired t-test. Data are shown as mean ± SD. Each point represents a single mouse (A-F). In A-F, one representative experiment of three independent experiments is shown.

however, a small amount can escape into the surrounding tissue, leading to extensive tissue damage and inflammation [34]. In addition, when accompanied by pathogenic infections, neutrophils and monocytes strongly induce the expression and secretion of S100a8/a9 to modulate inflammatory processes by inducing inflammatory cytokine production [35]. Real-time PCR analysis showed that MPO and S100a8 transcript levels were significantly higher in $Trim26^{-/-}$ kidneys than in $Trim26^{+/+}$ kidneys after *C. albicans* infection (Fig 3D). Additionally, we observed that the neutrophil infiltration into the kidneys of $Trim26^{-/-}$ mice was significantly higher than that in $Trim26^{+/+}$ mice (Figs 1D and 3A–3C). This may explain the increased expression of the proinflammatory cytokine $Trim26^{-/-}$ kidneys after fungal infection.

Neutrophils and macrophages play critical roles in innate antifungal immunity and function as the first line of defense against fungal infections [36]. After fungal infection, macrophages or DCs produce cytokines and chemokines, such as CXCL1 and granulocyte–macrophage colony-stimulating factor (GM-CSF), to recruit neutrophils and macrophages to the infected areas [37, 38]. Next, to determine whether the increased neutrophil infiltration in kidney of $Trim26^{-/-}$ mice was because of the up-regulation of chemokines, we explored the mRNA expression of the chemokines CXCL1 and CXCL2 in the kidney after *C. albicans* infection. We found that the transcription levels of CXCL1 and CXCL2 were strongly increased in $Trim26^{-/-}$ mice at 1–5 days post-infection (Fig 3E). Furthermore, the secretion of CXCL1 in the serum of $Trim26^{-/-}$ mice on day 5 post-infection was significantly higher than that in $Trim26^{+/+}$ mice (Fig 3F). To rule out the possibility that the difference in fungal load might induces a difference in CXCL1 expression and neutrophil recruitment, we examined these alterations when the fungal burden was similar. At 4 days post-infection, both the transcriptional level of CXCL1 in the kidney and the secretion of CXCL1 in the serum were higher in $Trim26^{-/-}$ mice (Fig 3G and 3H). However, the secretion of CXCL1 showed no significant different at early time points after infection (12 and 24 h) in $Trim26^{-/-}$ mice (S2D Fig). Neutrophil infiltration also increased significantly at 4 days post-infection, when the fungal burden was equivalent (Fig 3I). To further validate the role of CXCL1 in neutrophils recruitment and immunopathology in $Trim26^{-/-}$ mice, we used the CXCR1 and CXCR2 inhibitor, reparixin, to block the process of CXCL1 recruiting neutrophils. Indeed, using the *in vivo* model of invasive candidiasis, we found that reparixin administration at three consecutive doses at 2, 4, and 6 days after *C. albicans* infection resulted in significantly prolonged survival of $Trim26^{-/-}$ mice compared with treatment with the diluent alone (Fig 3J). We also observed decreased weight loss and reduced fungal burden in the kidneys of $Trim26^{-/-}$ mice treated with reparixin (Fig 3K and 3L). Administration of reparixin also reduced neutrophil recruitment in the kidneys of $Trim26^{-/-}$ mice, indicating that this inhibitor is effective (Fig 3M). These data indicated that Trim26 deficiency promotes renal inflammatory neutrophil recruitment by increasing CXCL1 production but not macrophages accumulation during candidiasis.

Th1 and Th17 cell differentiation play important roles in antifungal immunity. We further assessed whether Trim26 regulated Th1 and Th17 responses after *C. albicans* infection. We found that Th1 and Th17 levels were not markedly changed in the lymph nodes and spleen of Trim26-deficient mice compared with those in control mice after fungal infection (S3A and S3B Fig). Furthermore, we isolated spleen cells from infected $Trim26^{+/+}$ and $Trim26^{-/-}$ mice, and stimulated them with heat-killed *C. albicans* yeast *ex vivo*. Consistent with the *in vivo*

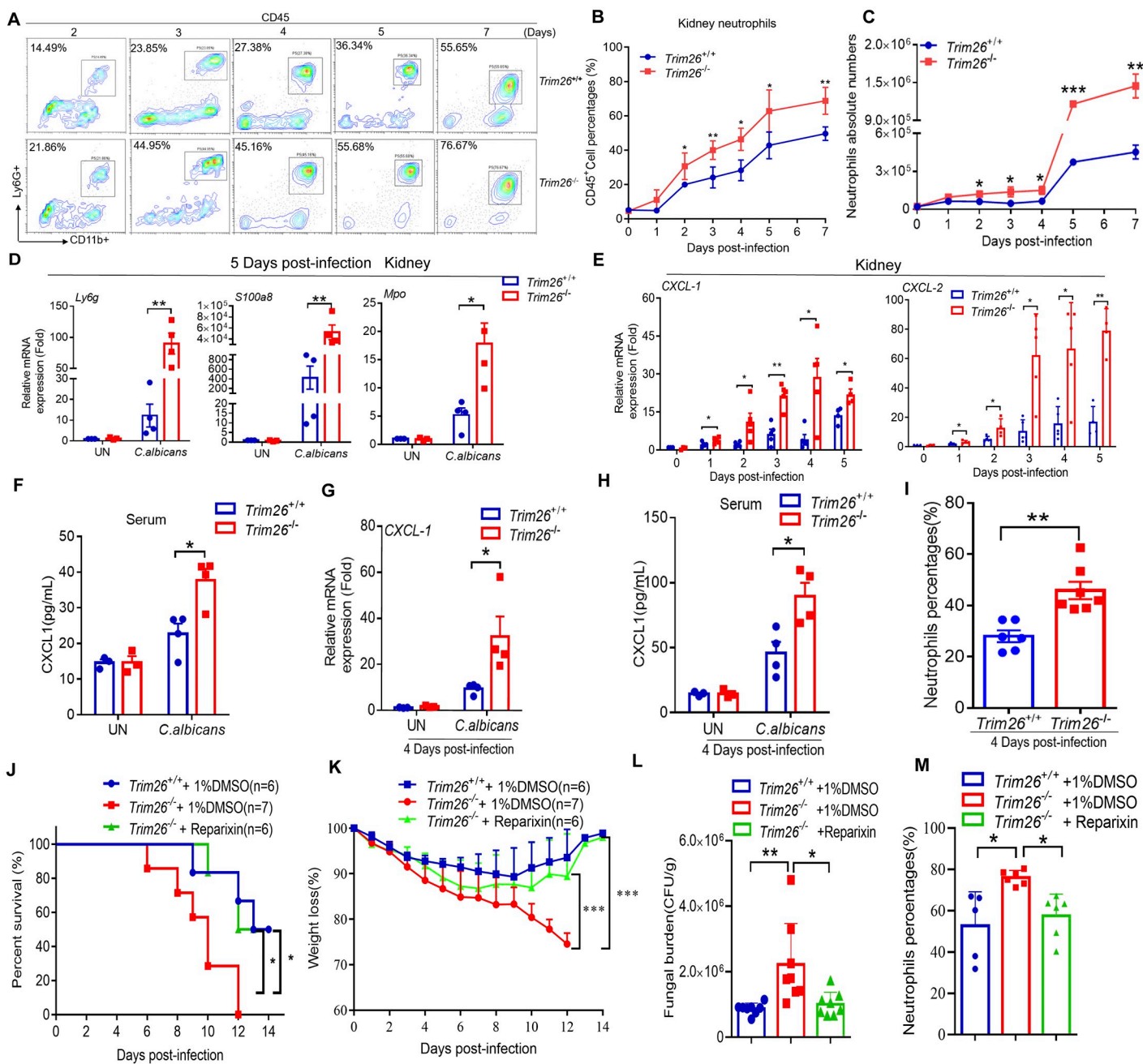

**Fig 3. Trim26 deficiency promotes renal inflammatory neutrophils infiltration via increased CXCL1 during candidiasis.** (A-C) WT mice or Trim26 KO mice were intravenously injected with live *C. albicans* ($1\times10^5$ cfu/100μL 1×PBS), and mice were euthanized at the indicated day after infection. Cells from kidneys were stained for flow cytometry analysis. Representative contour plot for $CD45^+Ly6G^+CD11b^+$ neutrophils (A) and their percentages or absolute numbers were presented (B and C, respectively). (D-F) WT mice or Trim26 KO mice were intravenously injected with live *C. albicans* ($1\times10^5$ cfu/100μL 1×PBS), and mice were euthanized at the indicated time. Mice were euthanized at day 5 after infection. Kidney was isolated and followed by real-time PCR analysis of indicated neutrophil-derived proteins expression (D). The mRNA expression level of indicated chemokines was determined by qPCR for Cxcl1 and Cxcl2 in kidneys (E). (F) Level of CXCL1 concentration in serum after 5 days of infection. (G-I) WT mice or Trim26 KO mice were intravenously injected with live *C. albicans* ($1\times10^5$ cfu/100μL 1×PBS), and mice were euthanized at four day after infection when fungal burden was identical. mRNA expression level of CXCL1 in kidney was determined by qPCR (G), level of CXCL1 concentration in serum was determined by ELISA (H) and neutrophils recruitment was determined by flow cytometry (I). (J-M) WT mice or Trim26 KO mice were intravenous injected with live *C. albicans* ($2\times10^5$ cfu/100μL 1×PBS), and Trim26 KO mice intraperitoneal injected with DMSO or Reparixin (10 μg/50μL) three consecutive doses on 2, 4, and 6 days after *C. albicans* infection. Survival rate (J) or weight loss rate (K) of mice was shown. (L-M) Mice were treated as in (K), and 7 days after *C. albicans* infection, mice were sacrificed, kidneys were lysed to count the fungal number by series plating (L) and neutrophils recruitment was determined by flow cytometry (M). *: P<0.05; **: P<0.01; ***: P<0.001 based on two-tailed unpaired t-test (B-I, L and M), Log-rank (Mantel-Cox) test (J) and two-way ANOVA (K). NS: no significance. Data are shown as mean ± SD. Each point represents a single mouse (D-I, L and M). In A, B, D, E, one representative experiment of three independent experiments is shown. In C, F-M, one representative experiment of two independent experiments is shown.

experiment, the percentages of Th1 and Th17 cells were were similar in between $Trim26^{+/+}$ and $Trim26^{-/-}$ spleens (S3C Fig). We also analyzed the abundance of B cells, CD4$^+$ T cells, and CD8$^+$ T cells in the kidneys and found no differences between $Trim26^{+/+}$ and $Trim26^{-/-}$ mice (S3D–S3F Fig).

## The renal infiltrated neutrophils of Trim26-deficient mice were found to have stronger inflammatory phenotype and impaired antifungal activity

To further investigate the mechanism by which Trim26 deficiency causes severe renal immuno-pathology and increases mortality during systemic *C. albicans* infection, we performed a tran-scriptomic analysis of neutrophils sorted from the kidneys of $Trim26^{+/+}$ and $Trim26^{-/-}$ mice seven days after *C. albicans* i.v. challenge (Fig 4A). We found 1703 differentially expressed genes, among which 1136 genes were upregulated and 567 genes were downregulated in $Trim26^{-/-}$ neutrophil compared to their $Trim26^{+/+}$ counterparts (Fig 4B). This was consistent with the findings that Trim26-deficient mice had elevated inflammatory damage of kidneys (Fig 2), Kyoto Encyclopedia of Genes and Genomes (KEGG) pathway analysis indicated that transcriptional networks related to inflammatory responses pathway were involved in the JAK-STAT signaling pathway, TNF signaling pathway, NF-κB signaling pathway, MAPK signaling pathway, and PI3-Akt signaling pathway (Fig 4C). In addition, we found that genes involved in inflammatory responses were significantly enriched in neutrophils from $Trim26^{-/-}$ mice com-pared with the corresponding gene-expression levels in $Trim26^{+/+}$ mice using Gene set enrich-ment analysis (GSEA) (Fig 4D). Furthermore, genes encoding inflammatory mediators, such as IL6, IL12a, IL12b, and CXCL1, were upregulated in $Trim26^{-/-}$ neutrophils (Fig 4E). Real-time PCR analysis confirmed that the levels of these inflammatory mediators were significantly higher in $Trim26^{-/-}$ neutrophils than in $Trim26^{+/+}$ neutrophils (Fig 4F). Collectively, these data suggested that Trim26 deficiency promotes the upregulation of inflammatory gene expression in kidney-infiltrated neutrophils during candidiasis.

Next, we investigated whether the inflammatory phenotype affects the antifungal activity of renal-infiltrated neutrophils in $Trim26^{-/-}$ mice. We sorted renal-infiltrated neutrophils from the kidneys by flow cytometry on 7 post-infection and compared their potential antifungal activity, including phagocytosis, neutrophil extracellular traps (NETs), reactive oxygen species (ROS) or nitric oxide (NO) production, and killing activity *ex vivo*. We found that the phago-cytic efficiency and NETs of $Trim26^{-/-}$ neutrophils did not differ from those of $Trim26^{+/+}$ neu-trophils (Figs 4G and S4A). However, the fungal-killing capacity of $Trim26^{-/-}$ neutrophils was markedly lower than that of $Trim26^{+/+}$ neutrophils (Fig 4H). Accordingly, ROS production, but not NO production, was significantly reduced in $Trim26^{-/-}$ neutrophils compared with that in $Trim26^{+/+}$ neutrophils (Figs 4I, 4J and S4B). Consistently, genes associated with ROS syn-thesis, such as Nox4 and Mmp8, were significantly downregulated in $Trim26^{-/-}$ neutrophils. Meanwhile, the genes associated with ROS clearance were significantly upregulated in $Trim26^{-/-}$ neutrophils compared with the corresponding gene-expression profiles in $Trim26^{+/+}$ neutrophils, including Cat, Mpo, and Sod2 (Fig 4K). *C. albicans* infection induces the recruit-ment of NK cells into the kidneys. NK cells produce GM-CSF to increase the phagocytic and fungicidal activities of neutrophils [39]. Therefore, we determined whether the change in GM-CSF could affect neutrophils' fungicidal activity and found no difference in GM-CSF pro-duction between $Trim26^{-/-}$ and $Trim26^{+/+}$ kidneys on days 3, 4, and 7 post-infection (S4C Fig). Together, these data indicated that renal-infiltrated neutrophils of Trim26-deficient mice have a stronger inflammatory phenotype and neutrophil-mediated kidney immunopathology, which accelerates the death of $Trim26^{-/-}$ mice. Additionally, Trim26-deficient renal-infiltrated neutrophils have impaired antifungal activity owing to reduced ROS production.

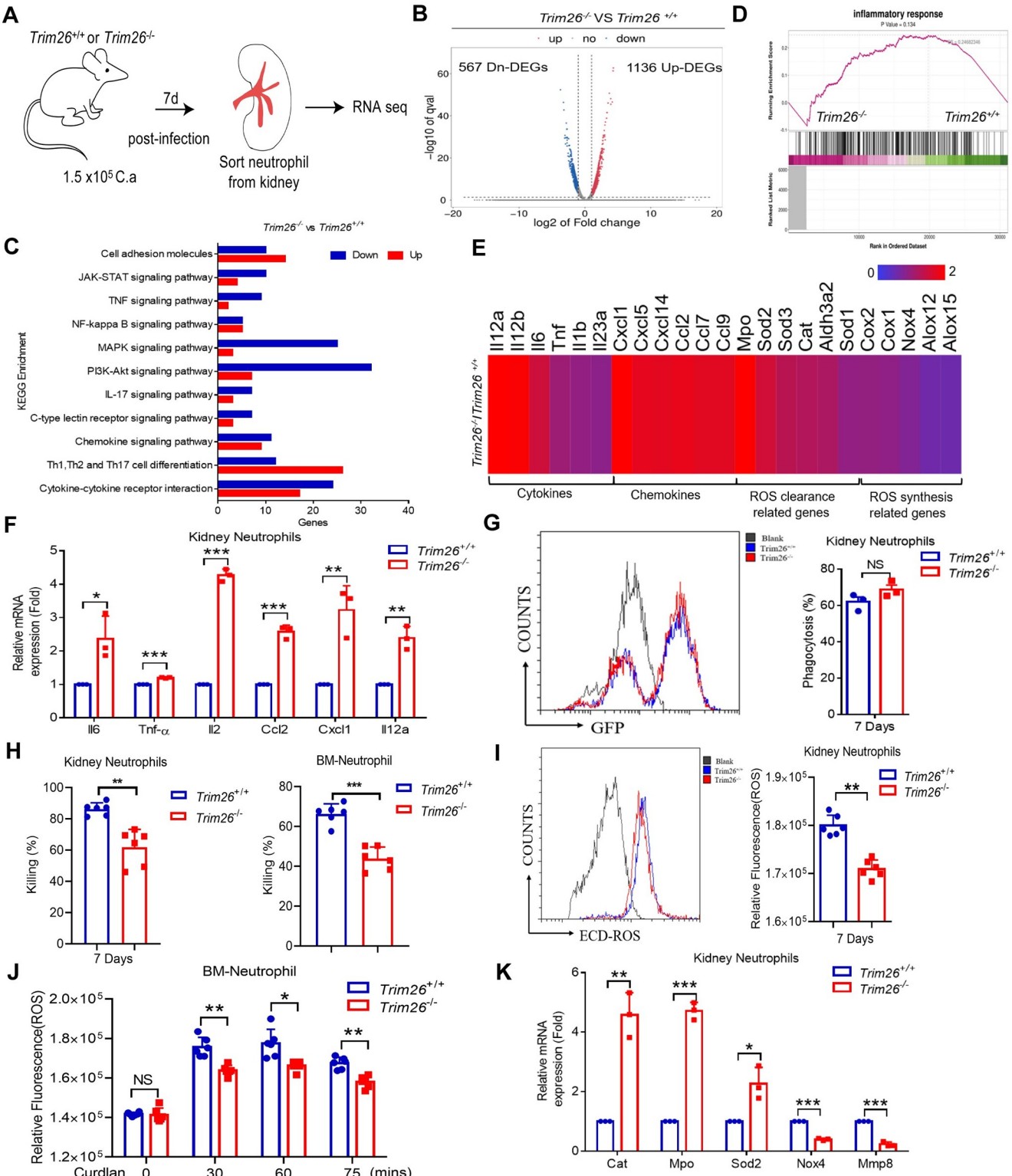

**Fig 4. Renal infiltrated neutrophils from Trim26-deficient mice had stronger inflammatory phenotype and impaired antifungal activity.** (A-E) Gene expression profiles controlled by Trim26 in kidney-infiltrating neutrophils from mice with systemic candidiasis. WT mice or Trim26 KO mice (n = 2 per genotype) were infected with live *C. albicans* (1×10$^5$ cfu/100μL 1×PBS), and mice were euthanized at day 7 after infection, kidney-infiltrating neutrophils (CD45$^+$Ly6G$^+$CD11b$^+$) were sorted for RNA-seq analyses. (A) Experimental scheme. Figure was illustrated using images created with Adobe Illustrator 2020 software. (B) Volcano plot describing the abundance of differentially expressed genes (DEGs). Two vertical dashed lines on each side correspond to

-1.0 and 1.0 cut-points, which are Log$_2$Fold Change cutoffs used for determining DEGs. The horizontal dashed line corresponds to P-adjusted values of 0.05, which was another cutoff used for determining DEGs. (C) Pathway enrichment analysis demonstrating up-regulation of various inflammatory pathways. (D) Gene set enrichment analysis (GSEA) of inflammatory responses genes. (E) Heat-map of select inflammatory genes and regulating ROS clearance and synthetic gene in cells was shown. (F) Neutrophils were isolated from kidney as the same as (A) and followed by real-time PCR analysis of indicated proinflammatory genes. (G) Phagocytosis of WT mice or Trim26-KO mice kidney-infiltrating neutrophils after infecting live *C. albicans* at 7 days. (H) Killing capacity of WT mice or Trim26-KO kidney-infiltrating neutrophils after infecting live *C. albicans* at 7 days or neutrophils from bone marrow as assessed by co-culture with *C. albicans*. (I) WT mice or Trim26-KO kidney neutrophils cells were stimulated heat-killed *C. albicans* (MOI = 10) for 30 min, followed by measurement of ROS production via a fluorescent ROS probe. (J) WT mice or Trim26-KO bone marrow neutrophils cells were stimulated curdlan (100 µg/mL) at the indicated time, followed by measurement of ROS production via a fluorescent ROS probe. (K) Neutrophils were isolated from kidney as the same as (A) and followed by real-time PCR analysis of indicated ROS synthesis related genes expression. *: P<0.05; **: P<0.01; ***: P<0.001 based on two-tailed unpaired t-test. NS: no significance. Data are shown as mean ± SD. Each point represents a single mouse (F-K), In F-K, one representative experiment of three independent experiments is shown.

## Trim26-deficient macrophages and DC cells promoted the production of CXCL1

After fungal infection, macrophages or DCs produce cytokines and chemokines, such as CXCL1 and GM-CSF to recruit neutrophils and macrophages to the infected area [37, 38]. To further explore why CXCL1 concentration was higher in *Trim26*$^{-/-}$ mice serum and kidney, we performed RNA-seq analysis of *Trim26*$^{+/+}$ and *Trim26*$^{-/-}$ BMDMs after stimulation with curdlan (100 µg/mL). We identified 356 upregulated transcripts and 425 downregulated transcripts following stimulation with curdlan (Fig 5A). KEGG pathway analysis revealed that the genes with the most changed expression profiles were involved in the immune system process, innate immune response, and C-type lectin receptor signaling pathway (Fig 5B). The chemokines and cytokines that were most affected by Trim26 deficiency were CXCL1, CXCL2, CXCL14, CCL3, IL-6, TNF, and IL-1a (Fig 5C). To confirm the RNA-seq data, we used real-time PCR to examine chemokine CXCL1 expression and found that CXCL1 expression induced by curdlan was significantly increased in Trim26-deficient BMDMs, BMDCs and renal macrophages compared with the corresponding CXCL1 expression level in the controls (Fig 5D and 5E). Consistently, CXCL1 chemokine secretion was also significantly increased in the Trim26-deficient BMDMs compared with the CXCL1 secretion in WT BMDMs after treatment with HKCA-Y, HKCA-H, D-zymosan, and α-mannan (Fig 5F). Together, these data indicated that Trim26-deficient BMDMs or BMDCs induce increased CXCL1 chemokine production upon fungal stimulation, which might contribute to higher levels of CXCL1 in the serum and kidney and increased neutrophils infiltration into the kidney.

To exclude the possibility that the excessive expression of chemokines in *Trim26*$^{-/-}$ macrophages was because of enhanced phagocytosis, we examined whether Trim26-deficient cells manifested any change in phagocytosis. We did not detect any differences in phagocytic uptake between *Trim26*$^{+/+}$ and *Trim26*$^{-/-}$ BMDMs, BMDCs or PMs (S5A–S5C Fig). The antifungal immune response can induce ROS to distinguish between bacterial and fungal propagation [40]. NO can also kill invading pathogens and interacts with ROS to form peroxynitrite to efficiently kill fungi [41, 42]. Next, we explored whether Trim26 deficiency affects ROS and NO production. Consistent with phagocytosis, we found that ROS or NO production was not impaired in Trim26-KO BMDMs compared with that in WT cells after treatment with HKCA-Y (S5D and S5E Fig). Consistently, *in vitro* fungal killing was not attenuated in Trim26-KO BMDMs, BMDCs and PMs compared with that in WT control cells (S5F Fig). These results demonstrated that Trim26 deficiency does not affect macrophages phagocytosis or fungicidal activity.

## Trim26 facilitated STAT1 nuclear translocation to inhibit CXCL1 expression

To further explore the mechanism by which Trim26 regulates the expression of CXCL1, we first examined the role of Trim26 in antifungal signaling in mouse BMDMs. We found that

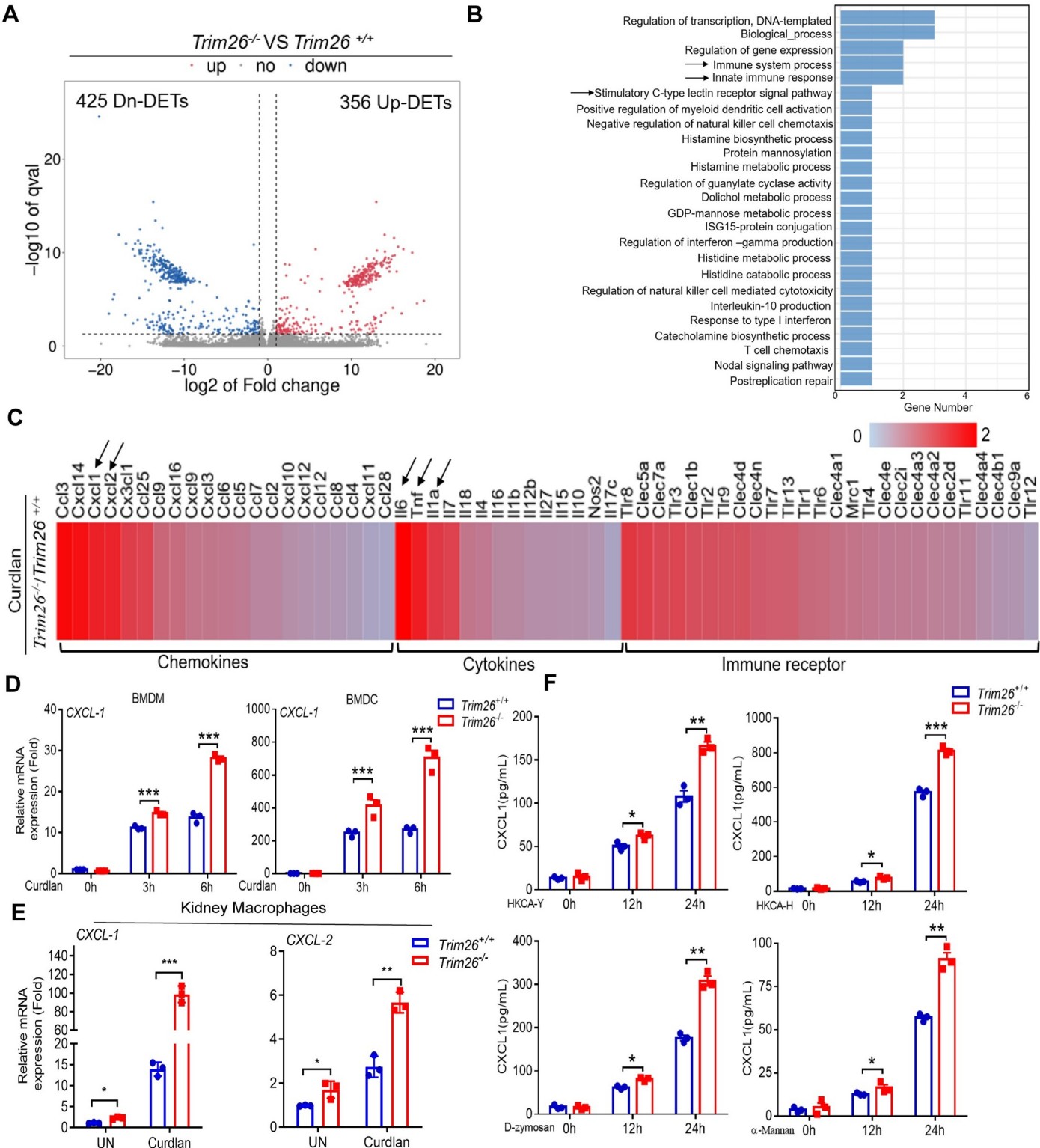

**Fig 5. Trim26-defficient macrophages and DC cells promoted production of CXCL1.** BMDMs from WT mice or Trim26 KO mice were stimulated with Curdlan (100μg/mL), followed by RNA-Seq analysis of gene expression. (A) Volcano plot describing the abundance of differentially expressed transcripts (DETs). (B) Pathway enrichment analysis demonstrating up-regulation of innate immune response pathways. (C) Heat-map of RNA-Seq data was shown. Arrow indicated top-changed gene expression in cytokine and chemokine groups. (D) BMDMs or BMDCs from WT mice or Trim26 KO mice were stimulated with curdlan (100 μg/mL) for the indicated times, followed by real-time PCR analysis of chemokine CXCL-1 expression. (E) Renal macrophages from WT mice or Trim26 KO mice were

stimulated with curdlan (100 μg/mL) for 6 hour, followed by real-time PCR analysis of chemokine CXCL-1 and CXCL-2 expression. (F) BMDMs from WT mice or Trim26 KO mice were stimulated with HKCA-Y (MOI = 5), HKCA-H (MOI = 5), D-zymosan (100 μg/mL) or α-mannan (100 μg/mL) for the indicated times, followed by ELISA analysis of CXCL1 production.*: P<0.05; **: P<0.01; ***: P<0.001 based on two-tailed unpaired t-test. NS: no significance. Data are shown as mean ± SD. Each point represents a single mouse (E), In D-F, one representative experiment of three independent experiments is shown.

the induction of NF-κB and MAPK signaling by heat-killed *C. albicans* or D-zymosan did not differ in the absence of Trim26. However, the level of p-STAT1 induced by the dectin-1 ligand D-zymosan or heat-killed *C. albicans* was dramatically reduced in Trim26-deficient BMDMs compared with that in BMDMs from WT mice (Fig 6A and 6B). Therefore, we proposed that Trim26 might interact with STAT1. We overexpressed the Myc-TRIM26 or (and) Flag-STAT1 plasmid in HEK293T cells and found that Trim26 could interacts with STAT1 (Fig 6C and 6D). Trim26 is an E3 ubiquitin ligase; therefore, we investigated whether Trim26 is involved in STAT1 ubiquitination. HEK293T cells were co-transfected with Flag-STAT1, HA-ubiquitin, and Myc-TRIM26 WT or TRIM26 (C16A), which lacks its enzymatic activity [28, 29]. We found that the polyubiquitination of STAT1 was readily detected in the presence of WT Trim26. However, TRIM26 (C16A) failed to do so (Fig 6E). Next, we investigated which type of STAT1 polyubiquitination is mediated by Trim26. We overexpressed the HA-ubiquitinmutants K48 and K63, which contained the one lysine residue at the indicated position, and the other lysine residues were substituted with arginine. As expected, Trim26 catalyzed the STAT1 polyubiquitination in the presence of WT ubiquitin. Furthermore, we found that Trim26 substantially increased STAT1 polyubiquitination in the presence of K63, but not with K48 ubiquitin mutants (Fig 6F), indicating that Trim26 mainly promotes K63-linked polyubiquitination of STAT1. To validate the function of the K63-linked polyubiquitination of STAT1, we isolated nuclear and cytosolic fractions from BMDMs after stimulation with HKCA-Y. We found that HKCA-Y stimulation induced STAT1 translocation to the nucleus. Notably, STAT1 nuclear translocation was greatly decreased in *Trim26*$^{-/-}$ BMDMs (Fig 6G). Consistent with our data, STAT1 has been reported to act as a negative regulator to inhibit CXCL1 expression [43, 44]. Therefore, Trim26-mediated K63-linked polyubiquitination and nuclear translocation of STAT1 may repress CXCL1 expression, and Trim26 deficiency relieves this inhibitory effect and enhances CXCL1 transcription and secretion.

## Discussion

Trim26 participates in various biological processes, including cell proliferation, autophagy, innate immunity, and inflammatory response [25–28]. Our previous study found that Trim26 negatively regulates IFN-β production and antiviral response through polyubiquitination and degradation of nuclear IRF3 [28]. We also found that Trim26 positively regulates the inflammatory immune response through K11-linked ubiquitination of TAB1 in the toll-like receptor signaling pathways [29]. In the present study, we investigated whether Trim26 plays an important role in the antifungal immune response. In a mouse *C. albicans* systemic infection model, we found that Trim26-deficient mice are more susceptible to *C. albicans*-induced fungal sepsis (Fig 1A). Further investigation showed that the excessive neutrophils infiltration and higher proinflammatory cytokines and chemokines production, which cause kidney damage, contributed to the higher mortality rate of Trim26-deficienet mice. Therefore, our study revealed a previously unrecognized function of Trim26 in antifungal immunity, which alleviates fatal immunopathology by regulating inflammatory neutrophils infiltration during *C. albicans* infection.

Neutrophils and macrophages play a critical role in antifungal innate immunity and function as the first line of defense to eliminate fungal infections [5, 36]. After a fungal infection,

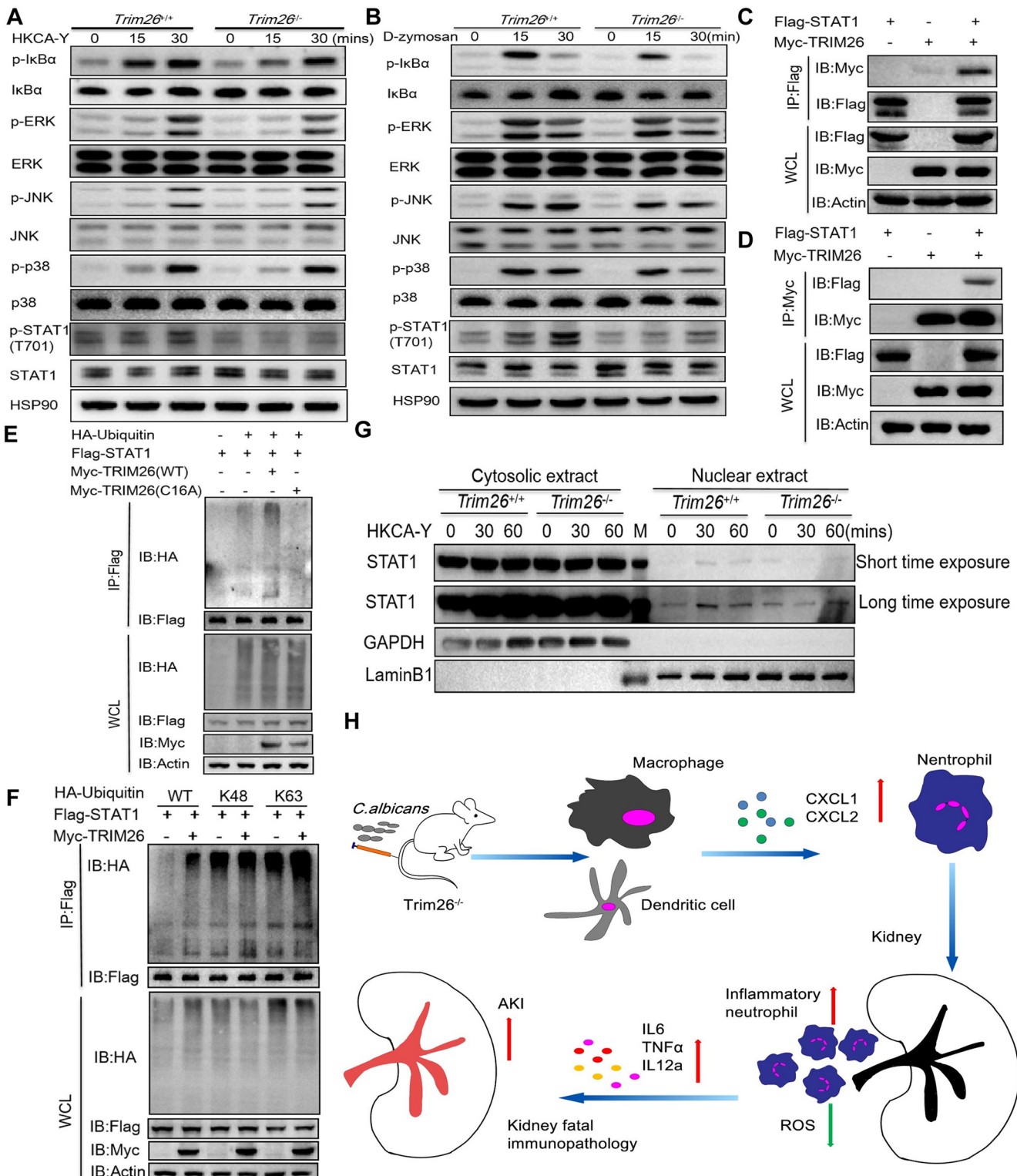

**Fig 6. Trim26 facilitates STAT1 nuclear translocation to inhibit CXCL1 expression.** (A-B) Western blot analysis of phosphorylated and total proteins in lysates of *Trim26*[+/+] and *Trim26*[−/−] BMDMs stimulated with HKCA-Y (MOI = 5) (A) or D-zymosan (B) for indicated time points. (C-D) HEK293T cells were transfected with indicated plasmids, and cell lysates were immunoprecipitated with anti-Flag antibody (C) or anti-Myc antibody (D), followed by immunoblot analysis for indicated proteins. (E) HEK293T cells were transfected with Flag-STAT1, HA-ubiquitin (WT), Myc-Trim26, or Myc-Trim26 (C16A) and ubiquitination assays were performed. (F) Ubiquitination assays analysis of STAT1 in HEK293T cells transfected with Flag-STAT1, Myc-Trim26, and HA-ubiquitin or its mutants (K48 and K63). (G) *Trim26*[+/+] and *Trim26*[−/−] BMDMs were stimulated with HKCA-Y for 30 or 60 min, and

whole-cell lysates were separated into nuclear and cytosolic fractions. Immunoblot analysis with indicated antibodies. (H) Schematic of the excessive inflammatory neutrophils infiltration leading to renal immunopathology in Trim26 KO mice. Figure (H) was illustrated using images created with Adobe Illustrator 2020 software. In A-G, one representative experiment of three independent experiments is shown.

macrophages or DCs produce cytokines and chemokines, such as CXCL1 and GM-CSF, to recruit neutrophils and macrophages to eliminate infections [37, 38]. Our data found that Trim26-deficient mice had excessive neutrophils infiltration into the kidneys during *C. albicans* infections. However, neutrophils do not promote fungal clearance, contradicting the conclusion that they play a protective role in antifungal immunity. RNA-seq analysis and the *in vitro* killing assay indicated that Trim26-deficient neutrophils exhibited a stronger inflammatory phenotype and impaired fungicidal activities (Fig 4F and 4H). However, the mechanism by which Trim26 regulates neutrophil hyperinflammation remains unknown and requires further investigation. Similar phenomena have been reported previously. For example, the neutrophil chemokine receptor Ccr1 amplifies late renal immunopathology and increases mortality in mouse systemic fungal infections by mediating the excessive recruitment of neutrophils from the blood to target organs [20]. IFN-I signaling controls the recruitment of inflammatory myeloid cells, including $Ly6C^{hi}$ monocytes and neutrophils, to infected mouse kidneys by driving the expression of the chemokines CCL2 and CXCL1, which aggravates renal immunopathology during *Candida* systemic infection [22]. In contrast, functional inactivation of suppressors of TCR signaling (Sts) enzymes leads to profound resistance to mouse systemic infection by *C. albicans* without the induction of hyperinflammatory responses in the kidney [23]. Functional alterations in neutrophils may be associated with the progression of *Candida* invasion. Significantly more neutrophils accumulate in the spleen and liver than in the kidneys during the first 24 h after infection, when the presence of neutrophil is critical for *Candida* infection control. Conversely, at later time points, only the kidney continues to accumulate neutrophils, which are associated with immunopathology and organ failure [21]. Therefore, the host immune system must precisely regulate the balance between effective host defense and immunopathology in invasive candidiasis.

Ubiquitination plays an important and extensive regulatory role in antifungal immunity. A recent study found that the E3 ubiquitin ligase CBLB controls proximal CLR signaling through associating with SYK and ubiquitinating SYK. CBLB deficiency results in increased inflammasome activation, enhanced ROS production, and increased fungal killing [45]. Another study found that CBLB is an E3 ubiquitin ligase for Dectin-1/2 that participates in antifungal immune regulation through the ubiquitination of Dectin-1/2[46]. These two studies suggest that CBLB deficiency or inactivation protects mice from systemic infection with a lethal dose of *C. albicans*, and CBLB may be a potential target for enhancing antifungal immunity. Our previous study revealed that the E3 ligase Trim31 is a crucial regulator of SYK activation. Trim31 interacts with SYK and catalyzes K27-linked polyubiquitination at Lys375 and Lys517 of SYK, promoting its plasma membrane translocation and binding with C-type lectin receptors. $Trim31^{-/-}$ mice were also more sensitive to *C. albicans* systemic infection than $Trim31^{+/+}$ mice [47]. We also found that the deubiquitinase OTUD1 directly interacts with CARD9 and cleaved polyubiquitin chains from CARD9, leading to the activation of the canonical NF-κB and MAPK pathways. Importantly, $Otud1^{-/-}$ mice are more susceptible to *C. albicans* systemic infection than wild-type mice *in vivo* [48]. Trim26 also belongs to the Trim protein family and exhibits E3 ligase activity. Our data show that Trim26-deficient macrophages and dendritic cells had higher levels of chemokines CXCL1 expression upon fungal stimulation. Trim26 deficiency inhibits K63-linked polyubiquitination of STAT1, and blocking STAT1 nuclear translocation may promote CXCL1 transcription and secretion.

In summary, we demonstrated the crucial role of Trim26 in antifungal responses by restricting inflammatory neutrophils infiltration and limiting proinflammatory cytokines production in the kidneys during *Candida* infections. These factors attenuate the fatal renal immunopathology and protect the host against fungal infections. Trim26-deficient neutrophils exhibit higher proinflammatory cytokine expression and impaired fungicidal activity. Excessive neutrophil infiltration in the kidney was because of the increased production of the chemokines CXCL1 and CXCL2 *in vivo* (Fig 6H). We found *Trim26*-deficient mice exhibit increased recruitment of neutrophils into the kidneys during the process of *Candida* infection (1–7 days post-infection). However, in the early stages of infection (1–3 days post-infection), neutrophils are protective against antifungal activity, with a lower fungal load in the kidneys. In the later stages of *Candida* infection, neutrophils show an inflammatory phenotype and have a decreased killing ability with increased fungal burden in the kidneys. How Trim26 affects neutrophil function at different stages of infection requires further investigation. Overall, our study elucidated the mechanisms by which Trim26 regulates inflammatory neutrophil infiltration and plays a protective role by alleviating renal injury during candidiasis.

## Materials and methods

### Ethics statement

All animal experiments were carried out following the general guidelines published by the Association for Assessment and Accreditation of Laboratory Animal Care. All of the mice were maintained under specific-pathogen-free conditions, and all animal studies were approved by the Scientific Investigation Board of School of Basic Medical Science, Shandong University (No. ECSBMSSDU2020-2-072).

### Mice

*Trim26*$^{-/-}$ mice on a C57BL/6 background were generated by Cyagen Biosciences Inc. (Guangzhou, China) using TALEN technology. The Trim26-deficient mice were genotyped by sequencing of a PCR fragment (250 bp) of the TALEN-targeting region amplified from genomic DNA isolated from tail tips using the following primers: forward 5′-GCTTGAGCAATC CACAGGGGTATC-3′ and reverse 5′-GCCACACGGGTCGGATGTTCT-3′. The depletion efficiency was confirmed by genotyping and western blot analysis.

### Plasmid constructs

Flag-STAT1 was generated by subcloning the human STAT1-coding sequence into the pCDNA3.1 vector. Myc-TRIM26, Myc-TRIM26 (C16A), HA-ubiquitin and other plasmids were described previously [28,29,47].

### Fungal strains and growth conditions

*C. albicans* strain SC5314 was provided by Dr. Changbin Chen (Institute Pasteur of Shanghai, CAS, Shanghai, China). *C. albicans* strain SC5314 was grown in peptone dextrose extract at 30˚C overnight, and aliquots were frozen at -80˚C. Yeast cells were cultured at 37˚C for 3 h in YPD medium plus 10% FBS to obtain hyphae. Yeast or hyphae cells were washed three times and resuspended in PBS buffer, then were incubated at 65˚C for 1 h to kill cells, then obtained HKCA-Y or HKCA-H. The death of heat-killed yeast or hyphae cells was verified by plating cells on YPD agar plates.

## Mouse BMDMs, BMDCs and PMs preparation

BMDMs or BMDCs were obtained by differentiating bone marrow progenitors from the tibia and femur of 6–8-week-old male or female mice in Iscove's Modified Dulbecco's Media (IMDM) containing 20 ng/mL of M-CSF, 10% heat-inactivated fetal bovine serum (FBS, Invitrogen), 1 mM sodium pyruvate, 100 U/mL penicillin, and 100 μg/mL streptomycin (Invitrogen) for 5–7 days. Cells were then re-plated in 6-well or 12-well plates 1 day before experiments. BMDCs were obtained by differentiating bone marrow progenitors from the tibia and femur of 6-8-week-old male or female mice in RPMI 1640 Media containing 20 ng/mL of GM-CSF and 10 ng/mL IL-4, 10% heat-inactivated fetal bovine serum (FBS, Invitrogen), 1mM sodium pyruvate, 100 U/mL penicillin, and 100 μg/mL streptomycin (Invitrogen) for 9 days. BMDCs were then re-plated in 6-well or 12-well plates 1 day before experiments. Peritoneal macrophages (PMs) were harvested from mice 3 d after thioglycollate injection and cultured in DMEM supplemented with 10% heat-inactivated fetal bovine serum (FBS, Invitrogen), 100 U/mL penicillin, and 100 μg/mL streptomycin (Invitrogen).

## Isolation of bone marrow neutrophils and kidney neutrophils

BM was flushed from femurs and tibiae of euthanized mice and passed through a 70-μm strainer using ice-cold PBS supplemented with 2% FBS and 2 mM EDTA. Neutrophils were isolated using MojoSort mouse Ly-6G selection kit (Biolegend, cat no. 480123). Neutrophils were resuspended in RPMI 1640 medium supplemented with 2 mM glutamine, 1% non-essential amino acids, 1% sodium pyruvate, 1% penicillin/streptomycin (all from Invitrogen), 10% FBS and incubated for the indicated time at 37˚C in a humidified atmosphere containing 5% $CO_2$. Cells were then re-plated in 96-well or 48-well plates 3 h before experiments.

For kidney neutrophils isolation and sorting, kidneys were perfused, minced, and placed in DMEM (Invitrogen) containing 400 mg/mL collagenase D (Roche, Mannheim, Germany) and 10 U/mL DNase I (Roche, Mannheim, Germany) for 30 min at 37 C. Digested kidney tissues were passed through a 40-μm cell strainer (BD Falcon). Subsequently, leukocytes were isolated by density gradient centrifugation. Cells were washed in PBS containing 2% BSA, and suspended in 36% Percoll (Merck Millipore, Darmstadt, Germany), and gently overlaid onto 72% Percoll. After centrifugation at 872× g at room temperature for 30 min, cells were retrieved from the Percoll interface and washed twice in DMEM and once with staining buffer (PBS containing 2% BSA and 0.1% sodium azide). $CD45^+Ly6G^+CD11b^+$ neutrophils were purified with fluorescence-activated cell sorting (MoFlo Astrios EQ, Beckman Coulter).

## Reagents

Antibodies of anti-Trim26 (clone A-7) and anti-GAPDH (clone 2E3-2E10) were bought from SANTA CRUZ BIOTECHNOLOGY (cat no. sc-393832 and sc-293335). Flow antibodies of anti-CD3(clone 17A2), anti-CD4(clone GK1.5), anti-CD8(clone 53–6.7), anti-IL-17A(clone TC11-18H10.1), anti-CD45 (clone 30-F11), anti-Ly6G (clone 1A8), anti-F4/80 (clone BM8), anti-CD11b (clone M1/70) and anti-CD19 (clone 6D5) were bought from Biolegend (cat no: 100204, 100406, 100758, 506908, 103108, 127608, 123115, 101212 and 115512). Antibody of anti-IFN-γ (clone XMG1.2) was bought from eBioscience (cat no.11-7311-82). Zombie Violet Fixable Viability kit was bought from Biolegend (cat no. 423114). D-zymosan and curdlan were bought from Invivogen (cat no. tlrl-zyd and tlrl-cud). α-Mannan was bought from Sigma (cat no. M3640). Recombinant mouse M-CSF, GM-CSF and IL-4 proteins were bought from Peprotech (cat no. 315–02, 315–03 and 214–14). Sytox Green was bought from Thermo Scientific (cat no. S7020). CXCR1 and CXCR2 inhibitor Reparixin was purchased from Selleck (cat no. S8640). Antibodies of anti-IkB(clone44D4), anti–p-IkB(Ser[32]), Anti-p44/42 MAPK (Erk1/

2), anti–p-p44/42 MAPK (Erk1/2) (Thr$^{202}$/Thr$^{204}$), anti–p-SAPK/JNK (Thr$^{183}$/Tyr$^{185}$), anti-SAPK/JNK, anti–p-p38 MAPK (Thr$^{180}$/Tyr$^{182}$), anti–p38 MAPK, anti–p-STAT1 (Tyr$^{701}$), and anti–STAT1 were bought from CST (cat no:4812, 2859, 4695, 4370, 4668, 9252, 9211, 8690, 9167, 14994). Antibody of anti-Lamin B1 was bought from Abways (cat no. AB0054).

## Immunoblot

Cell were harvested and lysed on ice in lysis buffer which containing 0.5% Triton X-100, 20 mM Hepes pH 7.4, 150 mM NaCl, 12.5 mM β-glycerophosphate, 1.5 mM $MgCl_2$, 10 mM NaF, 2 mM dithiothreitol, 1 mM sodium orthovanadate, 2 mM EGTA, 20 mM aprotinin, and 1 mM phenylmethylsulfonyl fluoride for 30 minutes, followed by centrifuging at 12,000 rpm for 15 minutes to extract clear lysates. The whole cell extracts were resolved on SDS-PAGE followed by immunoblotting with antibodies.

## RT and Real-time PCR

Total RNA was extracted from spinal cord with TRIzol according to the manufacturer's instructions. 1 μg total RNA for each sample was reverse transcribed using the SuperScript II Reverse Transcriptase from Thermo Fisher Scientific. The resulting complementary DNA was analyzed by real-time PCR using SYBR Green Real-Time PCR Master Mix. All gene expression results were expressed as arbitrary units relative to expression *Actb* or *GAPDH*.

## RNA-seq

For the neutrophils RNA-seq experiment, kidney-infiltrating neutrophils from mice were isolated from control or Trim26-KO mice with systemic candidiasis. After 7 days infection, neutrophils in kidney were harvested and RNA extracted by Trizol. Samples were sent to the LC-Bio Technologies (Hangzhou) Co., LTD. for RNA-seq analysis. For the data analysis, gene expression from Trim26-KO cells/gene expression from control cells was calculated as the fold (the gene expression of control cell was normalized as 1). For BMDMs RNA-seq experiment, BMDMs were isolated from WT mice or Trim26 KO mice (three mice each genotype, and cells from each mouse were cultured separately as replicates, and cells from each mouse were divided to two parts as untreated or curdlan-treated when stimulating with curdlan). Cells were left untreated or treated with curdlan (100μg/mL) for 6 h, and were harvested by Trizol. Samples were sent to the LC-Bio Technologies (Hangzhou) Co., LTD. company for RNA-seq analysis. For the data analysis, gene expression from curdlan-treated Trim26-KO cell/gene expression from curdlan-treated control cells was calculated as the fold (the gene expression of control cell was normalized as 1).

## Flow cytometry

After live *C. albicans* infection, mice were sacrificed at the indicated time after infection and perfused with 1×PBS. Kidneys, spleen and lymph nodes were homogenized in ice cold tissue grinders, filtered through a 70 μm cell strainer and the cells collected by centrifugation at 400 g for 5 min at 4˚C. Cells were resuspended and subjected to flow cytometry. Cell surface staining was done for 30 min at 4˚C. Intracellular staining was done using a fixation/permeabilization kit (Biolegend). Zombie Violet Fixable Viability kit (1:400; Biolengend) was added to exclude dead cells. Flow cytometry data analysis was performed on CytExpert.

## Bone marrow–chimeric mice

Six-week-old recipient mice were lethally irradiated by X-ray (550 rad×2), and $5×10^6$ BM leukocytes from donors were intravenously transferred to the mice. Chimeras were used for further experiments 7 weeks after the initial reconstitution.

## Phagocytosis assay, ROS production assay and NO production assay

For neutrophils, BMDMs or BMDCs phagocytosis assays, *C. albicans* (strain SC5314) cells were labeled with FITC in 100 mM HEPES buffer (pH 7.5). Neutrophils, BMDMs or BMDCs were co-cultured with labeled *C. albicans* at 37˚C for 45 mins. Fluorescence signals from fungal cells that were adherent to, but not phagocytosed by, phagocytes were quenched with trypan blue, and the rate of phagocytosis was assessed by flow cytometry. For GFP-*C. albicans*, funguses were fixed by 2% paraformaldehyde at room temperature for 30 mins, then were co-cultured with neutrophils, BMDMs or BMDCs for the indicated times. Unbound yeasts were gently washed by 1×PBS with five times; cells were analyzed by flow cytometry.

For ROS production assay, ROS was measured by DHE (Dihydroethidium) fluorescent probe according to the manufacturer's instructions. Briefly, $2.5×10^5$ bone marrow neutrophils or $0.5×10^6$ BMDM cells were washed with PBS twice. DHE was added to fresh medium at the final concentration of 1 μM and cells were incubated at 37˚C for 30 minutes before stimulated with heat-killed *C. albicans* (MOI = 5) or curdlan (100 μg/mL). ROS production was determined by flow cytometry.

For NO production assay, BMDMs or BM neutrophils were stimulated with heat-inactivated *C. albicans* (MOI = 10) or curdlan (100 μg/mL). NO production in culture supernatants was measured by a Nitric Oxide Assay Kit (S0021) from Beyotime Biotechnology based on Griess Reagent. NO concentration was calculated from the standard curve of $NaNO_2$ in accordance with the manufacturer's protocol.

## Fungal killing assay

For *in vitro* fungal killing assay, WT neutrophils or Trim26-KO neutrophils ($5×10^5$/well) were incubated with live *C. albicans* ($1 ×10^4$) in the wells of a 48-well plate and were kept for 60 min at 4˚C to allow the cells to 'settle'. BMDMs, BMDCs and PMs were plated in replicates at a density of $5 × 10^5$/well in 24-well plates on the day before the assay. Cells were washed three times with medium before the addition of live *C. albicans* at a ratio of 0.4 yeast per cell. Cells were allowed to interact for 45 min at 37˚C with *C. albicans*. After co-culture, cells were washed by PBS with three times, then resuspended in fresh medium containing Amphotericin B (Sigma Aldrich, V900919) at a final concentration of 30 μg/mL and cultured for 3 h at 37˚C. Cells were washed three times in PBS again, then lysed by the addition of 0.02% Triton X-100 (Sigma Aldrich, T8787) and 100 μL suspension was spread (1:1000 dilution) on YPD plates. After incubation at 37˚C for 24 h, killing was determined by counting the *Candida* colonies.

## Systemic *C. albicans* infection model

Live *C. albicans* strain SC 5314 ($2×10^5$ yeast cells in 0.1 ml of 1×PBS buffer) were injected intravenously into 6- to 8-week-old littermates of distinct genotypes. Infected mice were monitored daily for weight loss and survival. Fungal burden of kidneys was measured 5 days after infection. After the kidneys were collected, tissue homogenates were serially diluted and plated on yeast extract–peptone–dextrose agar. Fungal colony-forming units were counted 24 hours after plating.

## Histopathology

For histopathology analyses, kidneys were fixed in 10% neutral-buffered formalin, processed according to standard procedures, embedded in paraffin, and sectioned. 2-μm-thick sections were stained with hematoxylin and eosin (H&E), periodic-acid-Schiff (PAS) or anti-Ly6G (Servicebio; GB11229). Renal inflammation was scored based upon H&E and PAS staining (proportion of renal parenchyma and/or pelvis involved by tubulointerstitial nephritis and/or pyelonephritis) as not significant (score 0), less than 10% (score 1), 10–25% (score 2), 25–50% (score 3), or greater than 50% (score 4). The intra-lesional fungal burden was based upon PAS staining as not significant (score 0), scant presence in less than 10% of inflammatory foci (score 1), mild-to-moderate presence in 10–25% of inflammatory foci (score 2), moderate-to-significant presence in 25–50% of inflammatory foci (score 3), or significant presence in more than 50% of inflammatory foci (score 4). The extent of infiltration by Ly-6G$^+$ cells in affected kidneys was evaluated by Caseviewer software.

## ELISA

The concentrations of mouse CXCL1 from cell culture supernatants, serum or from kidney homogenates were determined by commercial ELISA Kits (Multisciences Biotech, China). The concentrations of mouse GM-CSF in kidney tissue homogenates were determined by ELISA Kits (Dakewe Biotech, China). All conditions were according to the manufacturer's instructions.

## Netosis assay

Purified BM neutrophils were added to 12-well plate ($2 \times 10^5$ cells/well) before stimulation with HKCA-Y (MOI 1:10). Four hours later, cells were fixed with 4% paraformaldehyde (Beyotime, China) for 20 mins, then PBS containing 10 μM Sytox green (Thermo Fisher) was added and cells were analyzed for NET release with high-speed confocal microscopy (Andor Dragonfly 200). Unstimulated neutrophils were used as controls.

## Statistics

Statistical significance between two groups was determined by unpaired two-tailed t test; multiple-group comparisons were performed using one-way ANOVA; the clinical scores and weight change curve were analyzed by two-way ANOVA for multiple comparisons. $P < 0.05$ was considered to be significant. Statistical analyzed data were expressed as mean ± SD. Biological replicates as indicated in the figure legend.

# Supporting information

**S1 Data. Excel spreadsheet containing, in separate sheets, the underlying numerical data and statistical analysis for Figure panels 1A, 1B, 1C, 1D, 1E, 1F, 2A, 2B, 2C, 2E, 2F, 3B, 3C, 3D, 3E, 3F, 3G, 3H, 3I, 3J, 3K, 3L, 3M, 4C, 4E, 4F, 4G, 4H, 4I, 4J, 4K, 5C, 5D, 5E, 5F, S1A, S1B, S1C, S1D, S1E, S1F, S1G, S1H, S2A, S2C, S2D, S3A, S3B, S3C, S3D, S3E, S3F, S4B, S4C, S5A, S5B, S5C, S5D, S5E, and S5F.**
(XLSX)

**S1 Fig. Characteristics of immune cells in the Trim26-defficient mice.** (A)The absolute cell number of immune organs were detected. (B-E) Cells were isolated from spleen (B), bone marrows (C), lymph nodes (D), and thymocytes (E) of WT mice or Trim26-KO mice, followed by flow cytometry analysis for the indicated immune cell populations. (F-H) Wild-type control

mice or Trim26-deficient mice were intravenous injected with live *C. albicans* ($2\times10^5$ cfu/100 μL 1×PBS), the fungal burden in the spleen, kidney and liver after 1, 2, 3, 4, 5 and 7 days post-infection were shown.*: $P<0.05$; **: $P<0.01$; ***: $P<0.001$ based on unpaired two-tailed t test (A-E and F-H). NS: no significance. Data are shown as mean ± SD. Each point represents a single mouse (A-E and F-H).In A, F-H, one representative experiment of three independent experiments is shown. In B-E, one representative experiment of two independent experiments is shown.
(TIF)

**S2 Fig. Trim26 deficiency does not impact neutrophils death and accumulation of macrophages during invasive candidiasis.** (A-D) WT mice or Trim26 KO mice were intravenously injected with live *C. albicans* ($1\times10^5$ cfu/100μL 1×PBS), and mice were euthanized at the indicated day after infection. Cells from kidneys were stained for flow cytometry analysis. (A) The death rate of neutrophils from kidneys were analysed by flow cytometry. (B) Representative contour plot for CD45⁺F4/80⁺CD11b⁺ macrophages and their percentages were presented (C). (D) The level of CXCL1 concentration in serum was determined by ELISA. *: $P<0.05$; **: $P<0.01$; ***: $P<0.001$ based on two-tailed unpaired t-test. NS: no significance. Data are shown as mean ± SD. In A-C, one representative experiment of three independent experiments is shown. Each point represents a single mouse (D). In D, one representative experiment of two independent experiments is shown.
(TIF)

**S3 Fig. Trim26 deficiency has no effect on Th1 and Th17 cell differentiation and infiltration of B cells and T cells in the kidney during invasive candidiasis.** (A-C) WT mice or Trim26 KO mice were intravenously injected with live *C. albicans* ($1\times10^5$ cfu/100μL 1×PBS), and mice were euthanized at day 7 after infection. Th1 and Th17 cells from lymph nodes (A) and spleen (B) were stained as indicated for flow cytometry analysis. (C) Splenic cells isolated from the WT mice or Trim26 KO mice infected with *C. albicans* ($1\times10^5$ cfu/100 μL 1×PBS) for 5 days, and stimulated with HKCA-Y (MOI = 10) for 2 days. Intracellular staining of Th1 (IFN-γ) and Th17 (IL-17A) were determined with flow cytometry. (D-F) WT mice or Trim26 KO mice were intravenously injected with live *C. albicans* ($1\times10^5$ cfu/100 μL 1×PBS), and mice were euthanized at day 7 after infection. B cells (D), CD4⁺ T cells (E) and CD8⁺ T cells (F) from kidney were stained as indicated for flow cytometry analysis. *: $P<0.05$; **: $P<0.01$; ***: $P<0.001$ based on two-tailed unpaired t-test. NS: no significance. Data are shown as mean ± SD. In A-F, each point represents a single mouse. In A-F, one representative experiment of three independent experiments is shown.
(TIF)

**S4 Fig. Trim26 deficiency does not impact neutrophil extracellular traps (NETs), NO production and GM-CSF production after *C. albicans* invasion.** (A) WT mice or Trim26-KO BM neutrophils cells were stimulated heat-killed *C. albicans* (MOI = 10), then the neutrophil extracellular traps of neutrophils was showed with staining Sytox Green. Scale bars, 10 μm. (C) WT mice or Trim26 KO mice were intravenously injected with live *C. albicans* ($1\times10^5$ cfu/100 μL 1×PBS), and mice were euthanized at the indicated day after infection. GM-CSF concentration in kidney was measured by ELISA analysis. (**B**) WT mice or Trim26-KO BM neutrophils cells were stimulated curdlan (100 μg/mL) or heat-killed *C. albicans* (MOI = 10) for the indicated time, followed by measurement of NO production via a Nitric Oxide Assay Kit. *: $P<0.05$; **: $P<0.01$; ***: $P<0.001$ based on two-tailed unpaired t-test. NS: no significance. Data are shown as mean ± SD. Each point represents a single mouse (B-C). In A, one representative experiment of two independent experiments is shown. In B-C, one representative

experiment of three independent experiments is shown.
(TIF)

**S5 Fig. Trim26 has no direct influence on phagocytosis and fungal killing of macrophages and DCs.** (A-C) Phagocytosis of WT mice or Trim26-KO mice BMDMs (A), BMDCs (B) or PMs (C) was evaluated by the method described in Materials and Methods. (D) WT mice or Trim26-KO BMDMs cells were stimulated heat-killed *C. albicans* (MOI = 10) for 30 min, followed by measurement of ROS production via a fluorescent ROS probe. (E) WT mice or Trim26-KO BMDMs were stimulated with heat-inactivated *C. albicans* (MOI = 10) for the indicated time. NO production in culture supernatants at the indicated time point was measured by nitrite assay kit. (F) Killing capacity of WT mice or Trim26-KO BMDMs, PM and BMDCs as assessed by co-culture with *C. albicans*. Assays were performed in triplicate. *: P<0.05; **: P<0.01; ***: P<0.001 based on two-tailed unpaired t-test. NS: no significance. Data are shown as mean ± SD. In A-F, one representative experiment of three independent experiments is shown.
(TIF)

**S1 Table. qPCR primers used in this study.**
(DOCX)

**S2 Table. The Entire RNA-Seq dataset including differentially expressed genes in kidney neutrophils, related to Fig 4.**
(XLSX)

**S3 Table. The Entire RNA-Seq dataset including differentially expressed genes in BMDMs, related to Fig 5.**
(XLSX)

## Acknowledgments

We thank Translational Medicine Core Facility of Shandong University for consultation and instrument availability that supported this work. We thank Han Wu and Qing Shui for assisting in performing western blot experiments at paper revision stage. We are also sincerely grateful to all lab members in Dr. Gao laboratory for their helpful discussion and suggestions.

## Author Contributions

**Conceptualization:** Wanwei Sun, Chengjiang Gao.

**Data curation:** Guimin Zhao, Wanwei Sun.

**Formal analysis:** Guimin Zhao, Yanqi Li, Tian Chen, Feng Liu, Yi Zheng, Bingyu Liu, Wei Zhao, Xiaopeng Qi, Wanwei Sun, Chengjiang Gao.

**Funding acquisition:** Guimin Zhao, Yi Zheng, Chengjiang Gao.

**Investigation:** Guimin Zhao, Yanqi Li, Wanwei Sun.

**Methodology:** Guimin Zhao, Yanqi Li, Tian Chen, Wanwei Sun.

**Project administration:** Guimin Zhao, Yanqi Li, Tian Chen, Feng Liu, Yi Zheng, Bingyu Liu, Wei Zhao, Xiaopeng Qi, Wanwei Sun, Chengjiang Gao.

**Resources:** Guimin Zhao, Yanqi Li, Tian Chen, Feng Liu, Yi Zheng, Bingyu Liu, Wei Zhao, Xiaopeng Qi, Wanwei Sun, Chengjiang Gao.

**Software:** Guimin Zhao, Wanwei Sun.

**Supervision:** Wanwei Sun, Chengjiang Gao.

**Validation:** Guimin Zhao, Yanqi Li.

**Visualization:** Guimin Zhao, Wanwei Sun.

**Writing – original draft:** Guimin Zhao, Wanwei Sun, Chengjiang Gao.

**Writing – review & editing:** Guimin Zhao, Tian Chen, Wanwei Sun, Chengjiang Gao.

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
