## [Decision Letter · Decision Letter 0]

8 Jul 2023

Dear Dr. Gao,

Thank you very much for submitting your manuscript "TRIM26 alleviates fatal immunopathology by regulating inflammatory neutrophils infiltration during Candida infections" for consideration at PLOS Pathogens. As with all papers reviewed by the journal, your manuscript was reviewed by members of the editorial board and by several independent reviewers. In light of the reviews (below this email), we would like to invite the resubmission of a significantly-revised version that takes into account the reviewers' comments.

The manuscript has been reviewed by three experts in the field, and the critiques are mixed. There are several concerns that need to be addressed.

1. As reviewer 1 has pointed out, many of the changes in chemokines and cytokines may be caused by differences in fungal burden. Therefore it is imperative to revisit these experiments and analyze the soluble mediators when fungal burdens are similar if not identical.

2. To validate the role of CXCL1, the authors should neutralize it or block it and assess the impact on fungal burden. That piece of data will confirm that CXCL1 is the cause of the changes in neutrophil attraction.

3. The authors need to determine how TRIM 26 regulates CXCL1.

4. The authors need to reconsider their hypothesis that immunopathology is the cause of the more severe disease in the kidneys. The increase in fungal burden may be a cue for exaggerated neutrophil influx. The authors need to discriminate between the effect of the burden versus heightened recruitment. Again, examining the neutrophil response at a time when the burdens are similar would be needed. Or, infecting mice with different numbers of fungi to equalize the burden and then examine the neutrophil response.

5. Perform additional experiments to identify the root cause of the killing defect by neutrophils. As mentioned by reviewer 3 and I agree, the differences in ROS are not striking. And an n=2 is underwhelming.

6. Instead of bone marrow macrophages, the authors need to examine CXCL1 by renal macrophages. That is key to understanding the organ-specific effect.

We cannot make any decision about publication until we have seen the revised manuscript and your response to the reviewers' comments. Your revised manuscript is also likely to be sent to reviewers for further evaluation.

Sincerely,

George Deepe, jr., MD

Guest Editor

PLOS Pathogens

Alex Andrianopoulos

Section Editor

PLOS Pathogens

Kasturi Haldar

Editor-in-Chief

PLOS Pathogens

orcid.org/0000-0001-5065-158X

Michael Malim

Editor-in-Chief

PLOS Pathogens

orcid.org/0000-0002-7699-2064

The manuscript has been reviewed by three experts in the field, and the critiques are mixed. There are several concerns that need to be addressed.

1. As reviewer 1 has pointed out, many of the changes in chemokines and cytokines may be caused by differences in fungal burden. Therefore it is imperative to revisit these experiments and analyze the soluble mediators when fungal burdens are similar if not identical.

2. To validate the role of CXCL1, the authors should neutralize it or block it and assess the impact on fungal burden. That piece of data will confirm that CXCL1 is the cause of the changes in neutrophil attraction.

3. The authors need to determine how TRIM 26 regulates CXCL1.

4. The authors need to reconsider their hypothesis that immunopathology is the cause of the more severe disease in the kidneys. The increase in fungal burden may be a cue for exaggerated neutrophil influx. The authors need to discriminate between the effect of the burden versus heightened recruitment. Again, examining the neutrophil response at a time when the burdens are similar would be needed. Or, infecting mice with different numbers of fungi to equalize the burden and then examine the neutrophil response.

5. Perform additional experiments to identify the root cause of the killing defect by neutrophils. As mentioned by reviewer 3 and I agree, the differences in ROS are not striking. And an n=2 is underwhelming.

6. Instead of bone marrow macrophages, the authors need to examine CXCL1 by renal macrophages. That is key to understanding the organ-specific effect.

Reviewer's Responses to Questions

**Part I - Summary**

Reviewer #1: The manuscript by Zhao and colleagues analyzed the function of Trim26 during systemic candidiasis. The authors found that Trim26-deficient mice are more susceptible to lethal infection. The authors show that Trim26 restricts immunopathology by restricting CXCL1 secretion in macrophages and DCs (in vitro) to prevent excessive neutrophil influx. Some additional experiments are required to support the author’s findings.

Reviewer #2: This manuscript examines the role of Trim26 in pathogenesis in a murine model of invasive candidiasis. The finding that Trim26 deficient animals have higher proinflammatory cytokines and neutrophil recruitment in the kidneys is interesting. This is further linked to CXCL1 and CXCL2 production. Bone marrow transplant studies add rigor. Appropriate controls are included.

Reviewer #3: In this paper, the authors examine the role of Trim26 in host defense against systemic candidiasis. the authors show that Trim26-deficient mice have increased mortality and fungal burden after infection associated with increased renal failure. the susceptibility maps to the hematopoietic compartment. the authors show increase neutrophil accumulation in the knockout kidneys associated with increase neutrophil chemokines and a small change in killing capacity in knockout neutrophils. Upon exposure of bone marrow mononuclear phagocytes to curdlan Trim26 KO cells produced more chemokines. Although the work is of interest, there are several concerns with the conclusions of the manuscript and the presented data as outlined below:

**Part II – Major Issues: Key Experiments Required for Acceptance**

Reviewer #1: -The authors provide expression data from several cytokines. However, increased fungal burden (Fig. 1C) results in many cases to increased inflammation. The time point when fungal burden was measured is not indicated in the figure legend. In this context, is CXCL1 expression increased in Trim26 KO mice early during infection (12/24 hours)?

-Based on in vivo and in vitro data the authors speculate that CXCL1 is the driver for increased neutrophil influx and immunopathology in the Trim26 KO mice. To verify this the authors should deplete CXCL1 during infection and verify the phenotypes.

-The authors show that Trim26 KO BMDMs/DCs increases CXCL1 expression. However, it is unclear how Trim26 negatively regulates/restricts CXCL1 expression. The authors show look into the signaling pathways (CARD9–BCL10–MALT1, NF-κB, and MAPK) to determine involvement.

Reviewer #2: The introduction should be updated to include references to original data (instead of reviews). For discussion of known immune responses, please clarify if these are for murine or human studies. Please clarify systemic infection versus mucosal candidiasis.

Include biological replicates in figure legends.

Explain why curdlan was used for studies.

Reviewer #3: 1-the authors state that the problem in the KO kidneys is excess immunopathology. Actually the data presented here support a model by which Trim26 is important for resistance to infection, not immunopahtology modulation. The KO kidneys have more fungal load and as a result of that there are more neutrophils and more renal damage. The primary problem with KO kidneys and mice is impaired host defense, not immunopathology. Related to this, the sentence in lines 86-88 "In this study, we uncover that Trim26 attenuates fatal immunopathology by restricting inflammatory neutrophils infiltration and limiting the production of proinflammatory cytokines in kidney during Candida infections" is not entirely accurate. Please rephrase.

2-related to the above, it is unclear how KO mice cannot defend well against infection. The authors claim that the neutrophils in the KO mice have impaired ROS and killing capacity. However, the ROS MFI difference is at the 10% range, which is quite trivial. The killing defect in renal neutrophils is also small and only based on n of 2 where statistics are performed (which is not appropriate). therefore it remains unclear why the KO mice cannot control the fungal load in the kidney, which would be very important to establish to better understand why Trim26 promotes protective immunity.

3-the authors conclude that increased Cxcl1/Cxcl2 in the KO kidneys is responsible for the greater neutrophils in the kidney. However, the authors do not present any direct evidence for that, rather they present an association of the increased chemokine concentration with increased neutrophils in the kidney. Related to this, the sentences in Lines 197-199 "These data indicate that Trim26 deficiency promotes renal inflammatory neutrophils recruitment via increasing CXCL1 production," is not supported by the data.

4-the observation that KO mononuclear phagocytes in the bone marrow produce more chemokines is of interest. However, does this also apply to renal macrophages? That would be important to show

5-in Fig 3A, the authors show % of neutrophils in the kidneys. How about absolute numbers?

6- there are many grammatical and typo errors in the paper. Please use the help of a native English speaking person to help ameliorate those errors.

**Part III – Minor Issues: Editorial and Data Presentation Modifications**

Reviewer #1: (No Response)

Reviewer #2: Grammatical editing is required.

Some of the figures are small and difficult to read (example 1D).

Line 108: Clarify if the immune cell profile was for uninfected animals.

The function of Trim26 (E3 ligase) should be mentioned in the abstract.

Reviewer #3: (No Response)

PLOS authors have the option to publish the peer review history of their article (what does this mean?). If published, this will include your full peer review and any attached files.

Reviewer #1: No

Reviewer #2: No

Reviewer #3: No
---

## [Decision Letter · Decision Letter 1]

31 Oct 2023

Dear Dr. Gao,

Thank you very much for submitting your manuscript "TRIM26 alleviates fatal immunopathology by regulating inflammatory neutrophils infiltration during Candida infections" for consideration at PLOS Pathogens. As with all papers reviewed by the journal, your manuscript was reviewed by members of the editorial board and by several independent reviewers. The reviewers appreciated the attention to an important topic. Based on the reviews, we are likely to accept this manuscript for publication, providing that you modify the manuscript according to the review recommendations.

There is not a consensus on the quality of the paper. One reviewer considers that the authors have addressed their prior concerns. A second reviewer strongly believes that while the new data do add to the quality of the paper, there are still gaps that need to be considered and addressed. It is felt and I agree that there needs to be re-writing of the manuscript to make it flow better. In addition, the authors need to add a paragraph regarding the limitations of their study. This will help focus the reader on what is missing and needed for future. work.

Sincerely,

George Deepe, jr., MD

Guest Editor

PLOS Pathogens

Alex Andrianopoulos

Section Editor

PLOS Pathogens

Kasturi Haldar

Editor-in-Chief

PLOS Pathogens

orcid.org/0000-0001-5065-158X

Michael Malim

Editor-in-Chief

PLOS Pathogens

orcid.org/0000-0002-7699-2064

There is not a consensus on the quality of the paper. One reviewer considers that the authors have addressed their prior concerns. A second reviewer strongly believes that while the new data do add to the quality of the paper, there are still gaps that need to be considered and addressed. It is felt and I agree that there needs to be re-writing of the manuscript to make it flow better. In addition, the authors need to add a paragraph regarding the limitations of their study. This will help focus the reader on what is missing and needed for future. work.

Reviewer Comments (if any, and for reference):

Reviewer's Responses to Questions

**Part I - Summary**

Reviewer #1: The authors have added new data supporting their findings, and responded to all my queries

Reviewer #3: The authors have performed additional experiments in response to the reviewers' comments. Some of the prior issues have been clarified, but some have remained unclear in the opinion of this reviewer.

**Part II – Major Issues: Key Experiments Required for Acceptance**

Reviewer #1: (No Response)

Reviewer #3: (No Response)

**Part III – Minor Issues: Editorial and Data Presentation Modifications**

Reviewer #1: (No Response)

Reviewer #3: (No Response)

PLOS authors have the option to publish the peer review history of their article (what does this mean?). If published, this will include your full peer review and any attached files.

Reviewer #1: No

Reviewer #3: No

Figure Files:

Data Requirements:

Reproducibility:

References:

---

## [Editor Report · Decision Letter 2]

14 Dec 2023

Dear Dr. Gao,

We are pleased to inform you that your manuscript 'TRIM26 alleviates fatal immunopathology by regulating inflammatory neutrophil infiltration during Candida  infection' has been provisionally accepted for publication in PLOS Pathogens.

Best regards,

George Deepe, jr., MD

Guest Editor

PLOS Pathogens

Alex Andrianopoulos

Section Editor

PLOS Pathogens

Kasturi Haldar

Editor-in-Chief

PLOS Pathogens

orcid.org/0000-0001-5065-158X

Michael Malim

Editor-in-Chief

PLOS Pathogens

orcid.org/0000-0002-7699-2064
---

## [Editor Report · Acceptance letter]

26 Dec 2023

Dear Dr. Gao,

We are delighted to inform you that your manuscript, "TRIM26 alleviates fatal immunopathology by regulating inflammatory neutrophil infiltration during Candida  infection," has been formally accepted for publication in PLOS Pathogens.

Best regards,

Michael Malim

Editor-in-Chief

PLOS Pathogens

orcid.org/0000-0002-7699-2064